# Nonreciprocal field theory for decision-making in multi-agent control systems

Andrea Lama ®[1], Mario di Bernardo ®[1,2] ✉ & Sabine. H. L. Klapp ®[3] ✉

Field theories for complex systems traditionally focus on collective behaviours emerging from simple, reciprocal pairwise interaction rules. However, many natural and artificial systems exhibit behaviours driven by microscopic decision-making processes that introduce both nonreciprocity and many-body interactions, challenging these conventional approaches. We develop a theoretical framework to incorporate decision-making into field theories using the shepherding problem from swarm robotics as a paradigmatic example of a multi-agent control system, where agents, the herders, must coordinate to confine another group of agents, the targets, within a prescribed region. By introducing continuous approximations of two key decision-making elements - target selection and trajectory planning - we derive field equations that capture the essential features of this distributed control problem. Our theory reveals that different decision-making strategies emerge at the continuum level, from average attraction to highly selective choices, and from undirected to goal-oriented motion, driving transitions between homogeneous and confined configurations. The resulting nonreciprocal field theory not only describes the shepherding problem but provides a general framework for incorporating decision-making into continuum theories of collective behaviour, with implications for applications ranging from robotic swarms to traffic and crowd management systems.

Decision-making is a fundamental process underlying the dynamics of many complex systems in science and technology. At the macroscopic level, the emergent collective behaviour of these systems is often shaped by decisions made by individual agents at the microscopic scale. Examples span from animal groups coordinating their movements[1,2], to autonomous vehicles navigating traffic[3], to robotic swarms performing distributed tasks[4,5] and decision processes in economics[6]. Often these processes involve a predefined control goal. Understanding how such microscopic decisions translate into macroscopic behaviour is crucial for the analysis, design and control of such systems[7]. A versatile, and in many contexts, highly efficient tool to study emergent behaviours on large length and time scales are continuum equations comprising a set of coupled partial different

equations (PDEs). These not only allow to elucidate the type of instabilities and emerging dynamical states but can also help to identify relevant mechanisms and parameters ranges in dynamical systems with complex microscopic interaction rules. Yet, the precise translation of microscopic decision rules into continuum descriptions remains a significant challenge in physics and control theory.

Traditional field theories for complex systems often focus on collective behaviours arising from simple, typically reciprocal pairwise interaction rules. Major examples from the physics community are critical phenomena and phase separation in fluid and magnetic systems[8,9] and, more recently, active[10] and bio-matter[11]. In the last years, much attention has been devoted to nonreciprocal generalizations[12,13] characterized by asymmetrical couplings occurring,

[1]Modeling and Engineering Risk and Complexity, Scuola Superiore Meridionale, Naples, Italy. [2]Department of Electrical Engineering and ICT, University of Naples Federico II, Naples, Italy. [3]Institute of Physics and Astronomy, Technische Universität Berlin, Berlin, Germany. ✉e-mail: mario.dibernardo@unina.it; sabine.klapp@tu-berlin.de

e.g., in heterogeneous bacterial systems[14–16], synthetic colloidal mixtures in nonequilibrium environments[17,18], macroscopic predator-prey systems[19], systems with vision cones[20,21], but also neural[22] and social[23,24] networks, and in quantum optics[25–29]. These theories have demonstrated intriguing collective dynamics[12,13,30–35], and unusual material properties[36]; however, they do not take into account crucial ingredients of decision-making.

Incorporating decision-making with predefined control objectives introduces distinct challenges: agents must make decisions based on local observations to achieve some desired control goal, a fundamental characteristic of distributed feedback control strategies[7]. Here, we demonstrate that the decision-making process induces inherent nonreciprocity and non-pairwise couplings into the system dynamics by explicitly incorporating control objectives, e.g. considering the positions of agents relative to their distance from the goal region. This presents challenges for conventional field-theoretic approaches. Here, we propose a framework to address these challenges, using the paradigmatic example of the *shepherding* control problem, where a group of agents, the herders, must coordinate themselves to control the collective dynamics of a second group of agents, the targets, in a desired way[37].

The shepherding task considered here exemplifies key features of distributed control problems: (i) reliance on local observations and (ii) real-time decision-making to achieve global objectives[38,39]. In our setup, we analyze a representative shepherding task where $N_H$ herders must collect and contain $N_T$ targets within a pre-assigned goal region, which we assume to be a circle centered at the origin (see Fig. 1). To focus on the fundamental aspects of decision-making, we assume that targets exhibit no intrinsic collective properties, such as cohesion, velocity alignment, or coherent response[40], which would otherwise simplify the task[41]. We model the targets as Brownian particles that experience a physical (and, thus, reciprocal) volume exclusion between themselves (and with the herders) through soft repulsion within a distance $\sigma$, and are repelled by herders within a distance $\lambda$ (see Methods).

The key challenge in solving the shepherding control problem lies in the distributed decision-making that determines the dynamics of the herders that must achieve a collective task through local observations. Each herder faces two fundamental decisions that mirror challenges in many real-world control scenarios[7,42]: what target(s) to influence, and how to influence them. We incorporate these decision-making elements into the dynamics of each herder $i$ through a control input, denoted $\mathbf{u}_i$ (see Methods). This control

input takes the form of a feedback term that determines the herder's action based on the relative positions of the observed targets with respect to the goal region.

Following established approaches[37–39], each herder $i$ selects one target with position $\mathbf{T}_i^*$ at each time; specifically, it chooses the target furthest from the origin (i.e., the goal region) within its circular sensing region of radius $\xi$. To facilitate the later derivation of a continuum description that explicitly incorporates decision-making, we employ an approximation of this selection mechanism that was first proposed in previous work by some of the authors[37]. Specifically, we express the position of the selected target $\mathbf{T}_i^*$ as a weighted average of the position of the targets within the sensing region of the herder $i$. The approximation reads

$$\mathbf{T}_i^* = \frac{\sum_{a \in N_{i,\xi}} e^{\gamma(|\mathbf{T}_a| - |\mathbf{H}_i|)} \mathbf{T}_a}{\sum_{a \in N_{i,\xi}} e^{\gamma(|\mathbf{T}_a| - |\mathbf{H}_i|)}}, \tag{1}$$

where $N_{i,\xi}$ represents the set of targets within the sensing region of the $i$-th herder, $\mathbf{T}_a$ and $\mathbf{H}_i$ are the two-dimensional (2D) Cartesian coordinates of targets and herders respectively, and $\gamma \geq 0$ is a parameter that controls the selection specificity.

We wish to highlight that the selection rule approximation given by Eq. (1) for continuum descriptions was first introduced by Lama et al.[37] but its application was limited to stationary target distributions, with no explicit form derived for the resulting continuum dynamics. Our work significantly extends the results presented therein by developing a complete and explicit theoretical framework for dynamic multi-agent systems with decision-making capabilities.

Also, notice that both the selection rule in Eq. (1) and the resulting control input, given in Eq. (2) below, incorporate not only pairwise distances between agents, but also each agent's position relative to the goal. The resulting control has, therefore, a three-body character. The specific expression in Eq. (1), known as a soft-max function and widely used in neural networks for output selection[43], enables continuous tuning of the selection rule: as $\gamma \to \infty$, it recovers the selection of the furthest target, while for $\gamma \to 0$, $\mathbf{T}_i^*$ approaches the center of mass of all observed targets.

Once a target is selected, the herder positions itself at a distance $\delta$ behind it to drive it towards the goal. While more sophisticated approaches exist[44,45], in our minimal shepherding model the dynamics of herder $i$ is subject to an instantaneous feedback control input that

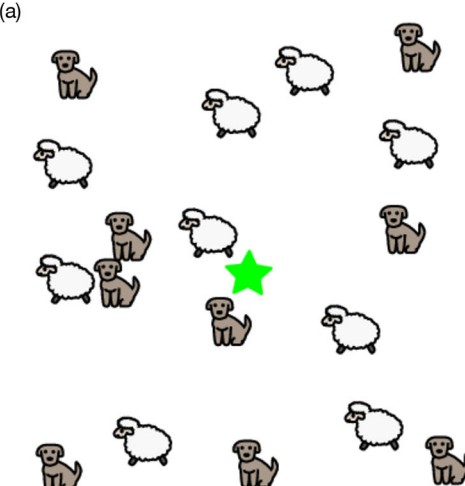

(a)

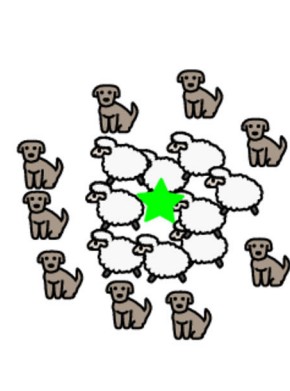

(b)

**Fig. 1 | Schematic illustration of the shepherding task. a** Initial random configuration of herders (dogs) and targets (sheep), with herders and targets randomly distributed in space and the goal region marked by a green star. **b** Final configuration showing successful containment of targets around the goal region (green star) by the coordinated action of the herders.

can be expressed as:

$$\mathbf{u}_i = - \left( \mathbf{H}_i - \mathbf{T}_i^* - \delta \widehat{\mathbf{T}}_i^* \right) \qquad (2)$$

where $\widehat{\mathbf{T}}_i^* = \mathbf{T}_i^* / |\mathbf{T}_i^*|$ is the unit vector pointing from the origin to $\mathbf{T}_i^*$. If no targets are within the sensing radius of herder $i$, then $\mathbf{u}_i = 0$. This feedback law linearly attracts the herder to the position $\mathbf{T}_i^* + \delta \widehat{\mathbf{T}}_i^*$, enabling it to push the target towards the origin when $\delta$ is smaller than the target's repulsion range $\lambda$. Similar to how $\gamma$ controls target selection, $\delta$ provides continuous control over trajectory planning: when $\delta = 0$, the herder approaches $\mathbf{T}_i^*$ from an arbitrary direction, generally failing to push it towards the origin, while for $\delta > 0$, the herder systematically drives the target towards the origin. Notice that also the trajectory planning, like the selection rule, has a three-body character, depending on the positions of the herder, the selected target, and the goal region.

These two parameters, $\gamma$ and $\delta$, thus enable us to continuously tune both aspects of decision-making in our system. As demonstrated later in Fig. 3, for sufficiently large $\gamma$ and appropriate values of $\delta$, the herders successfully confine the targets around the origin, achieving the shepherding goal. Conversely, when $\gamma = 0$ and $\delta = 0$, the system evolves to a homogeneous configuration (on average). As detailed in the Methods, our model includes white uncorrelated Gaussian noise for both herders and targets, as well as a reciprocal soft repulsion term between all agents within a distance $\sigma$, representing the agents' diameter, which we assume equal for herders and targets.

A key feature of our system, which will be fundamental in the derivation of the field equations, is its nonreciprocal nature. The dynamics violates the action-reaction principle of Newtonian mechanics, which is permissible since this principle only needs to hold at the microscopic level, not at the coarse-grained level of the agents[17]. While nonreciprocal effects have been extensively studied in physics[12,46], our model reveals a profound connection between nonreciprocity and decision-making: the very act of making decisions introduces fundamental asymmetries into the dynamics. Not only are targets repelled by herders while herders are attracted to targets, but herders also exhibit decision-making capabilities - selecting targets ($\gamma$) and choosing their positions with respect to a goal region ($\delta$) - that have no counterpart in the targets' dynamics. This intrinsic connection between decision-making and nonreciprocity suggests that nonreciprocal field theories are a natural framework for describing systems with distributed decision-making.

In this work, we show how the continuous tuning of decision-making rules through parameters $\gamma$ and $\delta$ enables the derivation of field equations that capture the macroscopic properties of the shepherding problem. Our analysis reveals that different decision-making strategies, from random to highly selective target choice and from undirected to goal-oriented pushing, emerge in the continuum description. The resulting nonreciprocal field theory is distinctly different from other nonreciprocal scalar theories such as the non-reciprocal Cahn-Hilliard (NRCH) theory for phase-separation in asymmetric systems[12,30,33] and predator-prey models[19]. Our approach does not only successfully describes the shepherding dynamics, including the transition from homogeneous to configurations where targets are confined, but also provides a general framework for incorporating decision-making in the presence of a pre-defined control goal into continuum theories of collective behaviour, allowing us to reproduce a variety of collective states. In our system, the predefined goal corresponds to a spatially fixed region at the center of the system. This results in an external field that breaks translational invariance: an aspect that, as shown in the paper, is of key importance for the structure of the resulting field equations. The presence of a predefined goal region is indeed a common feature in a variety of distributed control problems[47], from bacterial systems to robotic

swarm coordination[48], crowd management and autonomous transportation systems. As such, our framework opens new perspectives for analyzing and designing distributed control strategies across diverse fields of applications.

## Results

### A field theory for decision-driven shepherding

Our main result is a first-principles continuum theory that captures the essential physics of shepherding, derived from microscopic rules of agents' behavior to predict the emergence of macroscopic patterns. Specifically, we derive a mean field framework for shepherding dynamics based on two coupled partial differential equations (PDEs) that describe the spatio-temporal dynamics of the two species of herders and targets. Each species is represented by scalar, conserved density fields $\rho^A$, where A = H denotes herders and A = T denotes targets. The dynamics incorporate diffusion, conservative interactions, and, most importantly, decision-making elements. For simplicity, we focus on one-dimensional (1D) motion along the $x$-direction, though generalization to 2D is straightforward. To lowest order, the equations take the form (see Methods)

$$\partial_t \rho^T = \nabla \cdot \left[ D^T(\rho^T) \nabla \rho^T + \tilde{k}^T \rho^T \nabla \rho^H \right] \qquad (3)$$

$$\partial_t \rho^H = \nabla \cdot \left[ D^H(\rho^H) \nabla \rho^H - v_1(x) \rho^H \rho^T - v_2(x) \rho^H \nabla \rho^T \right] \qquad (4)$$

The intraspecies coupling in both equations is described by the renormalized diffusivities $D^A(\rho^A)$ which arise from noise and short-range agent-agent repulsion (see Methods). Additionally, the cross-coupling term with coefficient $\tilde{k}^T > 0$ in Eq. (3) combines the effects of both the reciprocal short-range repulsion and the nonreciprocal long-range repulsion exerted by the herders on the targets. This coupling results in the gradient of $\rho^H$ generating a current for $\rho^T$.

The decision-making capability of the herders is captured by the last two terms in Eq. (4) that involve the space-dependent functions $v_1(x; \gamma, \delta)$ and $v_2(x; \gamma, \delta)$. Examples of these functions are plotted in Fig. 2(a), while their analytical expressions are derived in the Methods. The role of the corresponding terms is sketched in Fig. 2(e–g). The function $v_2$, which multiplies the term $\rho^H \nabla \rho^T$, combines two contributions: the short-range reciprocal repulsion between herders and targets and the long-range attraction exerted on a herder by targets at position $x$. In our parameter regime, $v_2$ remains strictly positive, causing herders to move preferentially towards regions of higher target concentration [as shown moving rightward in Fig. 2(f)].

In contrast, function $v_1$, which multiplies the bilinear term $\rho^H \rho^T$, changes its sign at $x = 0$, the center of the goal region [indicated by a fence in Fig. 2(g)]. In particular, $v_1$ is negative for $x < 0$ and positive for $x > 0$. This change of sign reflects how the direction of chasing depends on the target's location relative to the goal. We may interpret $v_1$ as a task-dependent speed of herders when a non-zero concentration of targets $\rho^T$ is observed. The consequence is illustrated in Fig. 2(g). Despite observing a symmetric (homogeneous) distribution of targets, the herder preferentially chases the rightmost target due to its greater distance from the fence, representing the goal region. While our agent-based model yields explicit expressions for $v_1$ and $v_2$ (see Methods), we expect the overall dynamics to remain robust under small changes of these functions, provided their essential properties – particularly their signs and symmetries – are preserved.

Mathematically, the structure of the decision-making terms in Eq. (4), particularly the coupling term $v_1 \rho^T \rho^H$, emerges from the core properties of our agent-based model. These reflect how herders select which target to pursue (controlled by $\gamma$) and how they choose their chasing position (determined by $\delta$). In both decisions, herders consider the positions of the targets relative to their own positions and, most importantly, to the location of the goal region. The equations are

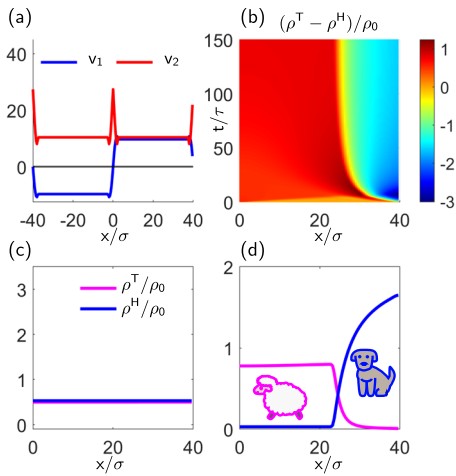

**Fig. 2 | Field theory description of shepherding dynamics. a** The coupling functions $v_1$ and $v_2$ generating shepherding dynamics; $v_2$ maintains a constant sign reflecting consistent attraction between species, whereas $v_1$ changes sign depending on the position $x$ of the herders with respect to the goal region $x = 0$, encoding goal-directed behaviour. **b** Spatiotemporal evolution of the density difference $\rho^T - \rho^H$: starting from a homogeneous distribution, the density profiles evolve and saturate to a shepherding configuration where $\rho^H$ effectively confines $\rho^T$ in a bounded region around the origin. (**c, d**) Steady state values of $\rho^T$ and $\rho^H$ for $\gamma = 0$, $\delta = 0$. **c** showing homogeneous distribution and for $\delta > 0$ and $\gamma > 0$. **d** Showing confined configuration. (**e–g**) Representative one-dimensional configurations of targets (sheep) and herders (dog) illustrating the corresponding nonreciprocal couplings in Eq. (4). **e** The herder observes a symmetric distribution of targets ($\nabla\rho^T = 0$), which generates no motion for the herder. **f** The herder observes an asymmetric distribution of targets ($\nabla\rho^T \neq 0$), and moves towards higher targets concentration; this is captured by the coupling $-v_2\rho^H\nabla\rho^T$. In (**g**), the target distribution is symmetric again ($\nabla\rho^T = 0$); however, the herder also possesses the additional information of the position of the goal region (fence) where to collect the targets. The herder will then move so as to complete the task; this is captured by the coupling term $-v_1(x)\rho^T\rho^H$ in Eq. (4).

derived by expanding the soft-max function in Eq. (1) for $\mathbf{T}_i^*$ in the small $\gamma$ regime, where $\gamma \lesssim 1/\xi$ (see Methods).

In the absence of decision-making ($\gamma = \delta = 0$), Eq. (3) remains unchanged, while Eq. (4) reduces to

$$\partial_t\rho^H = \nabla \cdot \left[ D^H(\rho^H)\nabla\rho^H - v_2^0\rho^H\nabla\rho^T \right] \quad (5)$$

where $v_1(x)|_{\gamma,\delta=0} = 0$, $v_2(x)|_{\gamma,\delta=0} = v_2^0$ being a positive constant, and the negative sign in front of the last term reflects the attractive nature of the force acting on a herder by a target (see Methods and Supplementary Information Section II.A for further details on the derivation). Even in this simplest case, the PDE system (3)–(5) is nonreciprocal in the sense that the cross-couplings between targets and herders have different signs (or values).

While similar scalar nonreciprocal theories, such as the non-reciprocal Cahn-Hilliard models, have been studied extensively[12,30,33] in relation to parity-time symmetry breaking and traveling states, our model exhibits a distinctive feature: for nonzero $\gamma$ and $\delta$, the current experienced by herders emerges from the control-oriented, three-body coupling that depends on the goal region's position. This three-body coupling gives rise to the final terms in Eq. (4). These terms have not only space-dependent prefactors, they also involve the bilinear nonreciprocal term $v_1(x)\rho^T\rho^H$, which is absent in previous theories without goal-oriented decision-making. The asymmetry between these terms and those in the simpler target dynamics in Eq. (3) constitutes a fundamental, source of nonreciprocity in the system, which remains unexplored at the field level.

### Field theory captures agent-based behaviour

Our field equations capture the complex spatial patterns and dynamics observed in agent-based simulations, validating the theoretical framework. To provide a microscopic perspective of the emerging shepherding dynamics, we present simulation snapshots from the underlying agent-based model (see Methods) in Fig. 3. We explore several representative values of the two control parameters: $\gamma$, which determines the selectivity, and $\delta$, which controls the trajectory planning, focusing on cases where the number of herders equals the number of targets ($N_H = N_T$) (for other cases, see Supplementary

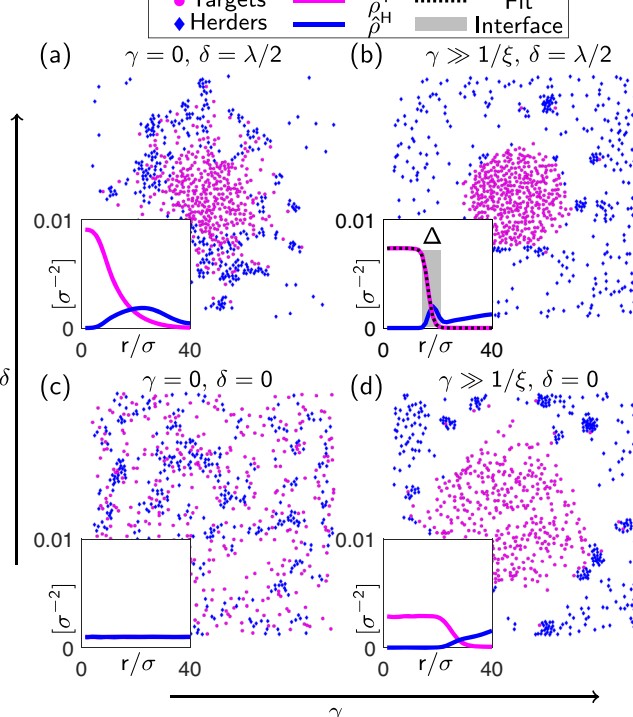

**Fig. 3 | Long-time configurations and density profiles of the agent-based system under varying control parameters. a–d** System states for different combinations of selectivity ($\gamma$) and trajectory planning ($\delta$) parameters, with insets showing time-averaged and angle-averaged density histograms of targets $\hat{\rho}^T(r)$ and herders $\hat{\rho}^H(r)$ as a function of the radial distance from the origin, $r$. At $\gamma = \delta = 0$ we observe a disordered state with (on average) homogeneous density distribution (**c**). The other panels show that, as soon as $\delta$ or $\gamma$ are non zero, inhomogeneous configurations are reached, where herders tend to surround targets. The inset of (**b**) shows the fit of the target profile to a hyperbolic tangent (black dashed line) and indicates with a shaded grey rectangle the target-herder interface of width $\Delta$, see Methods for further details.

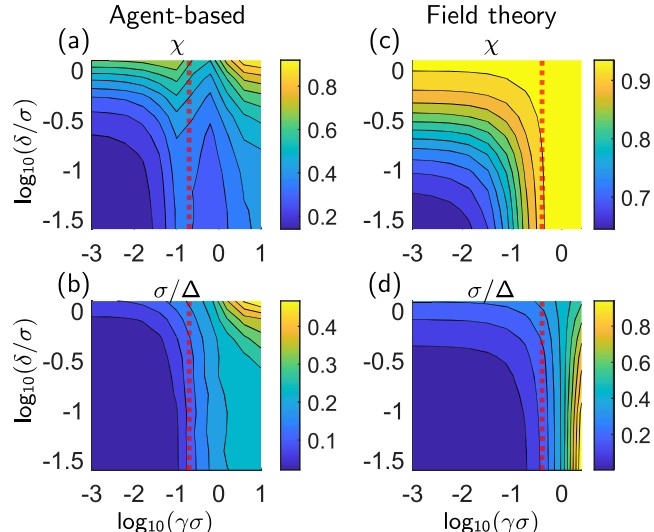

**Fig. 4 | Measures of shepherding performance at the microscopic and the continuum level.** Comparison of the values of $\chi$ (fraction of contained targets) and $\sigma/\Delta$ (sharpness of the herder-target interface) in the ($\gamma$, $\delta$) plane for (**a, b**) the agent-based model and (**c, d**) the field theory. For small values of $\gamma$, where the linearization made to derive the field equations is justified, the level curves show that the results of the agent-based model and of the field theories are consistent. Some differences appear around $\gamma \sim 1/\xi$ (red dashed line) where the non-linearity of the selection rule (1) becomes non negligible, hence where the linear approximation performed to derive the field equations starts to fail. A notable difference is that the combined maximum value of the two order parameters is reached for large $\gamma$ and $\delta$ for the agent-based model, and for large $\gamma$ and small $\delta$ on a field level.

Information Section I.C). The insets of Fig. 3 show the corresponding time-averaged density profiles relative to the origin, $\hat{\rho}^{\mathrm{T}}(r)$ and $\hat{\rho}^{\mathrm{H}}(r)$, where $r$ is the radial distance, obtained by averaging over the polar angle $\theta$.

In the absence of decision-making ($\gamma = \delta = 0$) the system reaches a disordered state [Fig. 3(c)] with homogeneous densities characterized by constant profiles. This behaviour is also observed at the continuum level (see Fig. 2(c)). Note that even in this uncontrolled case, the underlying equations of motion remain nonreciprocal. However, for the system parameters considered here, this nonreciprocity does not lead to overall symmetry breaking (see Supplementary Information Section II.B), but only induces microstructural changes (clustering) on the agent-based level, which are not the focus here and will be the subject of future investigation.

The situation changes when one or both of the control parameters $\gamma$, $\delta$ are nonzero: the system develops inhomogeneities, as illustrated by the density profiles in Fig. 3(a), (b), (d). This behaviour reflects a radial symmetry-breaking (relative to the case $\gamma = \delta = 0$), where the target density $\hat{\rho}^{\mathrm{T}}(r)$ reaches its maximum at the center and continuously decreases to zero radially outward. The most pronounced effect, which is key to obtain in real life scenarios such as in swarm robotics applications, occurs when $\gamma$ is large compared to the sensing radius and $\delta$ is non-zero, enabling herders to effectively guide their targets towards the desired direction. We then observe a concentrated disk of targets around the center, bounded by a sharp agglomeration of herders [cf. Fig. 3(b)].

Interestingly, even when only one control parameter is nonzero, similar but less pronounced inhomogeneities emerge [Fig. 3(a, d)]. When $\gamma = 0$ but $\delta > 0$, herders are attracted to the local center of mass of targets (rather than to a specific target), but approach them with a controlled distance strategy; this leads the herders to position themselves at the back of the barycenter of the observed targets, eventually creating an accumulation of herders around the targets. Conversely, when $\delta = 0$ but $\gamma > 0$, herders approach targets from random directions

but intelligently select optimal targets; this leads the herders to move towards the observed target with the largest distance from the origin, eventually generating a spatial inhomogeneity.

The continuum theory captures the transition from homogeneous to spatially structured states when decision-making is present. Two exemplary long-time sets of density profiles (from the numerical solution of Eq. (3) and (4)) are shown in Fig. 2(c, d), revealing the emergence of containment as we observed in their agent-based counterparts, despite differences in geometry (1D versus 2D) and specific values of $\gamma$ and $\delta$. Agent-based simulations in Supplementary Information Section I.D, where we consider a rectangular rather than circular goal region, further substantiate our claim that 1D field theories can capture the emergence of containment of the 2D agent-based dynamics. However, it is important to note that there are significant quantitative differences (for details, see the discussion in the next Paragraph). Indeed, in the absence of decision-making [Fig. 2(c)], the field theory produces a stable homogeneous state (note that temporal instabilities are absent at the parameters considered, see Supplementary Information Section II.B). With decision-making, the spatio-temporal evolution of the density difference $\rho^{\mathrm{T}}(x, t) - \rho^{\mathrm{H}}(x, t)$ in Fig. 2(b) clearly demonstrates the emergence of inhomogeneities and finally the saturation into a steady state with density profiles shown in Fig. 2(d). In this state, the targets are concentrated within the goal region around $x = 0$, confined by herders accumulating at larger distances, creating a sharp interface between these two spatial domains. This configuration matches the expected outcome of successful shepherding, see Fig. 1.

Importantly, the appearance of the inhomogeneous steady state is a direct consequence of the nonreciprocal coupling arising from decision-making in our field equations. A stable homogeneous steady state solution exists only in the absence of decision-making (i.e., $\gamma = \delta = 0$, $v_1 = 0$) (see Supplementary Information Section II.B). This solution ceases to be a steady state solution for any nonzero value of $\gamma$, $\delta$. Indeed, evaluated at the homogeneous solutions $\rho^{\mathrm{A}} = \rho_0^{\mathrm{A}}(t)$, Eq. (4) reads

$$\partial_t \rho_0^{\mathrm{H}}(t) = -\frac{d}{dx} v_1(x; \gamma, \delta) \rho_0^{\mathrm{T}} \rho_0^{\mathrm{H}}. \tag{6}$$

Thus, when $v_1$ becomes space-dependent (which occurs for nonzero $\gamma$, $\delta$), no homogeneous steady state exists. This holds at any order in the gradient expansion of the densities (see Supplementary Information Section II.A).

## Phase diagrams of shepherding states

Our analysis clearly reveals the emergence of inhomogeneous steady states whose details depend on two decision-making parameters. To quantify similarities and differences within these steady states, we analyze two key measures. First, we calculate the fraction $\chi$ of targets confined within a circle of radius $R$ around the origin, a metric commonly used to evaluate shepherding performance[37,45]. Here, $R$ corresponds to the position of the interface shown in Figs. 3(b), 2(d) for the continuum case. Second, we evaluate the width of the interfacial region, $\Delta$, where the density $\hat{\rho}^{\mathrm{T}}$ transitions from a non-zero value near the origin to zero at large distances. For details on the definition and calculation of $R$ and $\Delta$ in both agent-based and continuum descriptions, see Methods.

Results for both the levels of description for $\chi$ and the inverse width (sharpness) $\sigma/\Delta$ (where $\sigma$ is the repulsion distance representing the agents' physical size) as functions of the control parameters $\gamma$ and $\delta$ are shown in Fig. 4. Starting from the agent-based results [Fig. 4(a, b)], we see that both quantities approach their minimum value in the limits $\gamma \to 0$, $\delta \to 0$, as expected. The roles of $\gamma$ and $\delta$ are comparable for relatively small values, particularly when $\gamma \lesssim 1/\xi$ (red dashed line). Only beyond this value the nonlinearity of the selection rule given in Eq. (1)

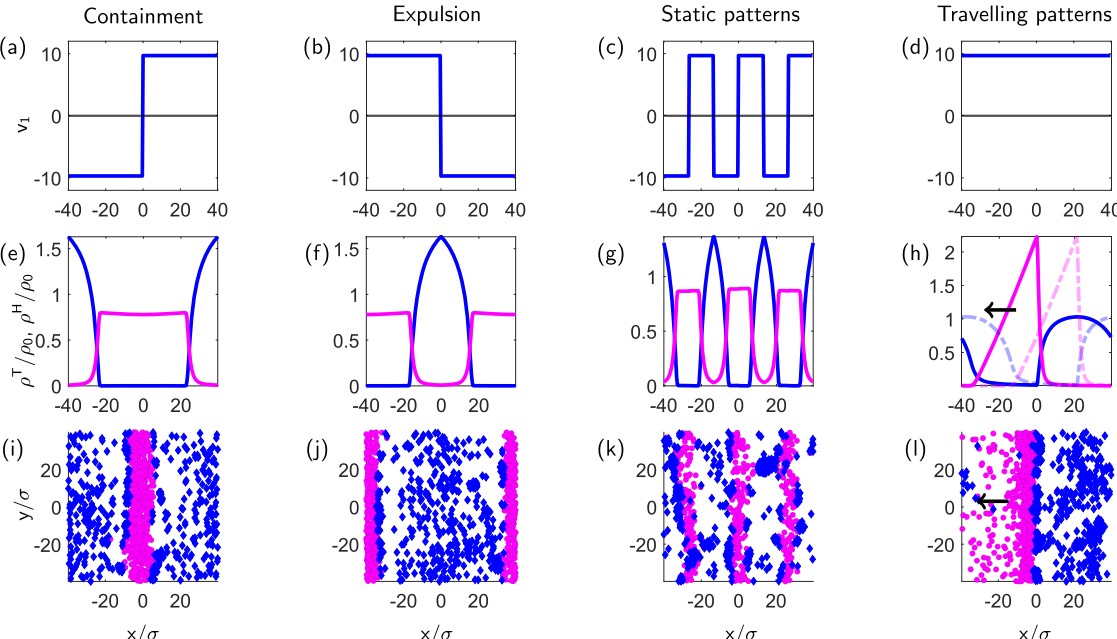

**Fig. 5 | Diverse collective state on the field level and at the agent-based level upon varying the control input and, thereby, the decision-making rules.** **a–d** Possible choices of the function $v_1(x)$, which we set as a square wave with different periods (**a–c**) or a constant (**d**). **e–h** The corresponding steady state behaviours of the field equations. (**i-l**) The corresponding steady state behaviours of the agent-based model; see Supplementary Information Section III.B for details on the agent-based and field equations. By suitable choosing $v_1(x)$, starting from a homogeneous state, we can achieve (**i, e**) confinement of the targets, as discussed above, (**f, j**) expulsion of the targets, e.g. the herders expel the targets out from a region around the origin, **g, k** static patterns, and (**h, l**) traveling patterns (see Supplementary Information Video 3 and 4.); in (**h**) we also plot in shaded dashed lines the densities at a previous time to show that the pattern is moving coherently in time.

becomes relevant. The level-sets of $\chi$, represented by $\delta(\gamma)$, show non-monotonic behaviour in $\gamma$ around this value. The largest values of $\chi$ and $\sigma/\Delta$ occur when both $\gamma$ and $\delta$ are large, confirming their roles as order parameters for the shepherding state[37].

For comparison, continuum results for $\chi$ and $\sigma/\Delta$ are shown in Fig. 4(c–d). As in the agent-based case, $\chi$ and $\sigma/\Delta$ reach their minimum values at $\gamma \to 0$, $\delta \to 0$. As $\gamma$ and $\delta$ increase from small values, $\chi$ increases monotonically with both parameters. This differs from the agent-based case [Fig. 4(a)], where the level sets of $\chi$, $\delta(\gamma)$, exhibit non-monotonic behaviour around $\gamma = 1/\xi$. This difference arises because the continuum equations employ a linearization with respect to $\gamma$ (see Methods), valid for $\gamma \lesssim 1/\xi$, a constraint absent in the agent-based case.

Notably, the sharpness reaches its maximum values at small values of $\delta$, a counterintuitive result explained by the competing roles of $v_1$ and $v_2$ at the continuum level (see Supplementary Information Section II.C). Hence, it is important to note that while our field theory predicts the absence of homogeneous steady states for nonzero $\gamma$ and/or $\delta$, and thereby the key qualitative feature of the shepherding dynamics, it does not fully capture all aspects of the spatial distributions observed in agent-based simulations. These discrepancies highlight limitations in our current framework that could be addressed in future refinements, possibly by incorporating higher-order terms in the expansion of the selection rule or modifying the mean-field assumptions.

### Design principles for decision-driven collective behaviour in natural and artificial systems

Our field theory framework reveals fundamental design principles for the emergence of decision-driven collective behaviour in multi-agent systems that extend beyond the basic containment task that we focused on so far. Through appropriate choice of the functions $v_1$ and $v_2$ in Eq. (4), we can generate a variety of controlled collective states that could serve as building blocks to capture behaviour observed in both natural and artificial engineered systems. These principles emerge more clearly when we examine how different choices of these coupling functions lead to distinct collective behaviours at long times, as illustrated in Fig. 5.

The function $v_1(x)$, which multiplies the product of density fields $\rho^T\rho^H$ in Eq. (4), plays a particularly crucial role as it encodes the task-dependent speed of herders. For simplicity, we set $v_1(x)$ to a square wave with different periods or phase shifts, while $v_2(x)$ is set to a positive constant (that, in this setting, represents an effective attraction of herders by targets). By choosing different spatial profiles for $v_1(x)$, we can generate (see Supplementary Videos):

- Containment: When $v_1(x)$ changes sign from positive to negative at $x = 0$, the targets remain confined to the goal area centered around the origin [Fig. 5(e)];
- Expulsion: Reversing the sign pattern, targets are expelled from a predefined region [Fig. 5(f)];
- Static patterns: When $v_1(x)$ alternates between positive and negative values periodically in space, stable, stationary stripe-like arrangements of targets are induced, [Fig. 5(g)];
- Traveling patterns: When $v_1(x)$ is set to a constant, non-zero value throughout space $v_1(x) = \bar{v}_1$, targets and herders create traveling waves that propagate through the system [Fig. 5(h)].

These examples indicate that already simple manipulations of the control coupling functions that encode basic rules for how agents make decisions, can induce a range of behaviours. For example, changes of sign of $v_1(x)$ and the location of the corresponding zeros can be used to realize desired static patterns without changing other coupling parameters. Such patterns are of interest, e.g., in swarm robotics[49] and in engineered bacterial populations[50,51]. Moreover, already the simplest choice for $v_1(x)$, namely a constant value, generates a traveling pattern (see Supplementary Information Section III.C for further details). We note that stripe-like and travelling patterns also emerge from other PDEs well established in the physics literature, such

as reaction-diffusion systems[52] and variants of the nonreciprocal Cahn-Hilliard model studied recently[12,30,33].

The key insight presented here is that, in decision-making systems, such diverse behaviours can emerge solely from varying decision-making rules of the herders. For example, we can generate travelling patterns in a parameter regime where such patterns are absent without decision-making (see Supplementary Information Section III.C). This perspective is further substantiated by agent-based simulation results (in the case of a rectangular goal region) shown in the bottom row of Fig. 5. To generate these results we did not change the microscopic interactions between the two types of agents, but only adapted the decision-making rules of the herders (see Supplementary Information Section III.B), using the same two-step strategy as in the shepherding problem: a proper selection rule, and a proper trajectory planning that finally determines $v_1(x)$ (see Supplementary Information Section III.B for further details). The fact that we can create such different collective behaviours by only modifying $v_1(x)$ suggests a unifying principle that may be common to both natural and engineered systems. Most notably, these behaviours emerge naturally from our field equations without requiring changes to their fundamental structure, indicating that our framework captures essential features of decision-driven collective behaviour that extend beyond the specific task we considered.

## Discussion

Our analysis uncovers a direct connection between microscopic decision-making and collective behaviour: agents' decision-making capabilities inherently generate both nonreciprocity and, in presence of a pre-defined goal, specific types of coupling in their continuum description. This led us to discover a class of nonreciprocal interactions that has remained unexplored in the related physics literature[12,30].

The coupling we derive, which originates from feedback control at the microscopic level (2) and involves a goal that, in essence, breaks translational invariance, manifests as shepherding behaviour at the continuum scale [Fig. 2(b)]. These results provide a mathematical framework connecting individual decision-making to collective behaviour, unifying concepts from physics and control theory. In the field of shepherding, that we have chosen as a paradigmatic example, our continuum framework can be systematically extended and refined: For example, it would be very easy to incorporate other types of microscopic interactions such as, e.g., cohesion or alignment, that would lead to additional couplings of the form $\nabla \cdot [\rho^A \nabla \rho^B]$ familiar from Cahn-Hillard-like theories. Furthermore, applications in higher dimensions could reveal additional instabilities. Indeed, given recent experiments in human coordination[38], we expect a polar instability induced by oscillatory motion of herders around targets. Another conceptually interesting question concerns changes imposed by considering nonlinear terms in the expansion of the selection rule.

The mathematical framework we developed here, where agents select whom to influence and how based on some control goal, can have implications beyond the specific shepherding task considered. Similar decision-making processes might play a role in biological systems, from marine predators[53] and killer whales[54] containing fish schools for hunting, to terrestrial cases like chimpanzee packs defending territories[55] and ant colonies coordinating excavation[56]. Similar principles might be present in social systems as well, particularly in opinion dynamics,[57] where influencers shape followers' opinion consensus by targeting certain individuals[58], and extend to human coordination[38,59], or engineered systems like swarm robotics[37,60], where artificial agents can be designed according to this paradigm. However, experimental validation would be needed to determine whether these natural systems actually employ similar mechanisms, an interesting open problem left for future study.

From a more conceptual perspective, our framework addresses an open challenge at the frontier between control theory and complex systems: steering collective behaviour in large-scale multi-agent systems to solve distributed control tasks. While advances have been made in controlling complex networks[7], distributed control of mobile agents faces unique challenges. Recent approaches in the field of active matter span optimal control[61,62], interaction design[20], feedback mechanisms[63], and machine learning[64,65]. Technically, at the agent level, two fundamental obstacles persist: the scaling of the number of dynamical equations with agent numbers and the time-varying nature of interaction matrices[66,67]. These challenges are particularly acute in shepherding, where the control of the targets is achieved indirectly through herder dynamics[68].

Here, we propose to tackle these challenges in complex, multi-agent control systems by deriving a continuum macroscopic description where the functions $v_1(x)$ and $v_2(x)$ emerge as natural control inputs for shaping collective behaviour. This aligns with growing interest from both physics[69] and control-theory communities to derive suitable partial differential equations that capture collective dynamics while incorporating precise control goals[70]. The resulting PDEs provide insights that hold across different system sizes and are insensitive to specific microscopic initial conditions - which are often impossible to prescribe in real-world applications anyway.

The generality of our framework, is necessarily bounded by the specific hypotheses made in our microscopic model and derivation. Clearly, there are a number of problems in which decision-making occurs without broken translational invariance, such as in consensus finding[71], pattern formation in swarms[49], and various socio- and economic problems that have been very recently analyzed through the lens of a field theory[72,73]. Still, our framework, which incorporates decision-making on the continuum level through an appropriately defined feedback current, may have relevance also for these more general decision-making problems.

In the present work we have provided some first examples of how deliberate manipulation of the control input $v_1$ and $v_2$ leads to a variety of collective behaviours. Future research could focus on application of machine learning techniques to actually learn the functions $v_1$ and $v_2$ directly from experimental data of natural or artificial systems exhibiting decision-driven collective behaviour. Similar strategies have been recently used to predict suitable PDEs for active and biological matter[69]. A related intriguing question is to which extent such learned functions could provide new insight into microscopic decision rules and interactions. The framework could also be extended to incorporate more sophisticated decision-making mechanisms, such as those including memory effects or adaptive learning. Furthermore, investigating the interplay between noise, decision-making, and collective dynamics could reveal dynamical phases and transition mechanisms. Ultimately, such PDEs should predict not only steady-state behaviour but also spatiotemporal patterns and even hierarchical self-organisation as it was recently found in biological communicating systems[74]. The connection between nonreciprocity and decision-making uncovered here also suggests new ways to design and control collective behaviour in synthetic systems, promising both deeper understanding of decision-driven collective phenomena and new possibilities for engineering complex systems with desired behaviours.

## Methods
### Agent-based model
The agent-based model underlying the field equations (3) and (4) was chosen as a modified version of the minimal shepherding model considered in earlier work by some of the authors[37]. It involves all of the essential ingredients for shepherding dynamics and is, at the same time, particularly suitable for the derivation of field equations.

Specifically, we consider a binary mixture of mobile, disk-like agents of type T or H (with 2D Cartesian position vectors $\mathbf{T}_a$ and $\mathbf{H}_i$,

respectively) moving in a square box of side length $L$ with periodic boundary conditions. We discuss here the case of a circular goal region around the origin; the case with a rectangular goal region, essentially representing a 1D problem, is discussed in Supplementary Information Section I.D.

The positions evolve in time according to overdamped Langevin dynamics, that is, both species are subject to Gaussian white noises representing the effects of (e.g., environmental or heterogeneity-induced) disorder on the coarse-grained scale (similar approaches were taken, e.g., in swarm robotics[75,76], collective behaviours in biology[77,78], and in crowd dynamics systems[79]). Further reasoning particularly for herder's noise is outlined in Supplementary Information Section I.A. We neglect inertial effects as it is often done in the robotics and control literature on shepherding[39,80]. The targets' and herders' dynamics are given by

$$\dot{\mathbf{T}}_a = k^{SR}\left(\sum_{b \neq a}^{N_T}\mathbf{F}_{SR}^{rep}(\mathbf{T}_a, \mathbf{T}_b) + \sum_{i=1}^{N_H}\mathbf{F}_{SR}^{rep}(\mathbf{T}_a, \mathbf{H}_i)\right)$$
$$+ k^T \sum_{i=1}^{N_H}\mathbf{F}_{LR}^{rep}(\mathbf{T}_a, \mathbf{H}_i) + \sqrt{2D}\mathcal{N}_a \tag{7}$$

and

$$\dot{\mathbf{H}}_i = k^{SR}\left(\sum_{j \neq i}^{N_H}\mathbf{F}_{SR}^{rep}(\mathbf{H}_i, \mathbf{H}_j) + \sum_{a=1}^{N_T}\mathbf{F}_{SR}^{rep}(\mathbf{H}_i, \mathbf{T}_a)\right) + k^H\mathbf{u}_i + \sqrt{2D}\mathcal{N}_i \tag{8}$$

respectively, where $\mathcal{N}_{a,i}$ are zero-mean white (i.e., delta-correlated) noises with unit variances, and the diffusion constants $D$ are assumed to be equal. All agents interact through symmetric (reciprocal) short-range (SR) repulsive forces $\mathbf{F}_{SR}^{rep}$ of amplitudes $k^{SR} > 0$ and range $\sigma$, the latter representing the (uniform) particle diameter. In addition, the targets [see (7)] are subject to a long-range (LR) repulsive force $\mathbf{F}_{LR}^{rep}$ from the herders, with amplitude $k^T > 0$ and range $\lambda > \sigma$. The LR repulsion reflects the tendency of a target to move away from an approaching herder. For both, SR and LR types of repulsion, as is common in the Literature (e.g.,[81,82]), we assume a linearly decreasing dependence on the distance, such that, e.g.,

$$\mathbf{F}_{LR}^{rep}(\mathbf{T}_a, \mathbf{H}_i) = \frac{\mathbf{d}_{ai}}{d_{ai}}(\lambda - d_{ai})\Theta(\lambda - d_{ai}) \tag{9}$$

where $d_{ai} = |\mathbf{d}_{ai}| = |\mathbf{T}_a - \mathbf{H}_i|$, and $\Theta(x) = 1$ if $x > 0$, 0 otherwise. This linear ansatz is particularly useful for our later derivation of field equations[12]. For the SR repulsion we have the same functional form of the interaction, but the range $\lambda$ is replaced by the particle diameter $\sigma$.

Beyond SR repulsion, the equation of the herders (8) contains the essential ingredient to realize shepherding, that is, the feedback control input $\mathbf{u}_i$ defined in Eq. (2). As described in the Introduction, the feedback term represents the (linear) attraction of a herder towards a strategic position behind its selected target. Specifically, this position is given by $\mathbf{T}_i^* + \delta\widehat{\mathbf{T}}_i$, where $\mathbf{T}_i^*$ [see Eq. (1)] is the position of the selected target (within a sensing radius $\xi$ and with selectivity $\gamma$), and $\delta\widehat{\mathbf{T}}_i$ represents a displacement along the target's position vector ($\widehat{\mathbf{T}}_i$ being the unit vector along $\mathbf{T}_i^*$). The parameter $\delta$ determines how far behind the target the herder should position itself to effectively steer the target's dynamics towards the origin. (As long as $0 < \delta < \lambda$ the herder can successfully navigate the selected target towards the origin.)

Equations (7), (8), combined with Eqs. (1) and (2), constitute the fundamental components of our agent-based model. A distinctive feature of these equations is that the coupling between targets and herders cannot be derived from a Hamiltonian and exhibits strong nonreciprocity in two key aspects: first, there is an inherent asymmetry in the interaction forces – targets experience repulsion from herders while herders are attracted to targets, which appears to violate Newton's third law (though this apparent contradiction is resolved when considering the coarse-grained nature of our description). Second, the selective nature of the herders' attraction introduces another form of nonreciprocity – herders interact only with specific targets chosen according to their selection rule and guide them toward the goal, whereas targets respond uniformly to all herders without any such selectivity.

To achieve effective confinement of targets (see Fig. 1), we assume that the sensing radius of the herders exceeds that of target-herder repulsion ($\xi > \lambda$), consistent with earlier literature[37,39,45]. Details of the simulations are provided next.

## Numerical calculations based on the agent-based model

**Technical details.** All simulations were carried out in MATLAB using an Euler-Maruyama integration scheme. The simulations are performed in a square periodic box of size $L \times L$ with $L = 80\sigma$, where $\sigma = 1$ defines the particle diameter. The diffusion constant is set to $D = 1$, which characterizes the strength of noise in the system. Using the characteristic time scale $\tau = \sigma^2/D$, we employ a time step of $dt = 1 \cdot 10^{-3}\tau$. For initial conditions, we randomly and uniformly distribute $N_H = 400$ herders and $N_T = 400$ targets in the simulation box, resulting in equal initial densities of $\rho_0^T = \rho_0^H = 0.0625/\sigma^2$ for both agent types (see Supplementary Information Section I.B for further details). The system typically reaches stationary states after a transient period of $t_{tran} = 200\tau$, and we run the simulations for a total duration of $t = 300\tau$.

Results for different values of the ratio $N_H/N_T$ are reported in Supplementary Information Section I.C.

The agent-agent interactions are characterized by three distinct force contributions, each with specific parameters. For the short-range repulsive forces, which act between all agents regardless of type, we choose the particle diameter as $\sigma = 1$ and amplitude $k^{SR} = 100/\tau$. The long-range target-herder repulsion is characterized by an interaction radius $\lambda = 2.5\sigma$ and amplitude $k^T = 3/\tau$. The long-range herder attraction has a sensing radius $\xi = 5\sigma$ and force amplitude $k^H = 3/\tau$. The snapshots in Fig. 5(i–l) are obtained with the same numerical values, and with $\gamma = 5/\sigma$ and $\delta = \lambda/2$; the dynamical equations are presented and discussed in Supplementary Information Section III.A.

**Density profiles and order parameters.** The radial density profiles $\hat{\rho}^T(r)$ and $\hat{\rho}^H(r)$ are computed by constructing histograms of the spatial distribution of targets and herders, respectively, as a function of their radial distance $r$ from the origin. These profiles are obtained by time-averaging over an interval of $t_{avg} = 100\tau$ after the system has reached a steady state.

The radial densities are computed using an array of equispaced points from the origin to $L/2$ (half the box length), with a step size of $\Delta r = \xi/4$. The normalization of these density profiles follows the condition $\int_0^{L/2} \hat{\rho}(r)r\, dr = \chi(L/2)$, where $\chi(L/2)$ represents the fraction of targets contained within a circle of radius $L/2$ centered at the origin. The profiles $\hat{\rho}^T(r)$ and $\hat{\rho}^H(r)$ are obtained by averaging over the polar angle $\theta$, which is appropriate given our focus on the radial structure of the confined state. Final results are obtained by averaging over an ensemble of 48 independent simulations.

We now define the two order parameters $\chi$ and $\sigma/\Delta$ considered in Fig. 4. First, to retrieve information on the width of the target-herder interface, $\Delta$, we fit the calculated density profile of the targets to a hyperbolic profile $\rho_f(r)$. This choice is inspired by theoretical studies on the vapor-liquid interface in phase-separating fluids[83]. The explicit expression for $\rho_f(r)$ is given by

$$\rho_f(r) = \rho_f(r; c, R, \Delta) = \frac{c}{2}\left(1 - \tanh\left(\frac{x - R}{\Delta}\right)\right) \tag{10}$$

where $c$, $R$, $\Delta$ are the fitting parameters representing, respectively, the maximum values of the profile, the position of the interface, and its width. We use the inverse of the width, $\sigma/\Delta$, as an estimate of the sharpness of the target-herder interface.

The confinement order parameter $\chi(R)$ measures the fraction of targets located within a distance $R$ from the origin. In Fig. 4, the reference radius $R=R^*$ is chosen as the fitting parameter $R$ of the density profile obtained at maximum values of $\gamma$ and $\delta$, where the spatial segregation between targets and herders is most distinct. The quantity $\chi$ is then calculated by averaging the instantaneous fraction of targets inside a circle of radius $R$, both over time ($t_{\text{avg}} = 100\tau$) and over an ensemble of 48 independent simulations. In Supplementary Information Section I.E we show that different choices of $R^*$ do not impact the qualitative behaviour of our results.

### Derivation of the field equations

Starting from the agent-based stochastic equations (7), (8) we now derive corresponding partial differential equations (PDEs) describing the spatial-temporal dynamics of the two density fields $\rho^A$, with A = H,T. The final structure of these equations is given in Eqs. (3) and (4). As in the Results Section we consider 1-dimensional dynamics along the $x$-direction; the derivation can be easily generalized to higher dimensions.

Due to the conservation of the number of particles, both density fields, defined as $\rho^{T(H)}(x,t) = \sum_{a(i)=1}^{N^{T(H)}} \delta\left(x - x_{a(i)}^{T(H)}(t)\right)$ where $x_a^T = T_a$ ($x_i^H = H_i$) are the 1D Cartesian coordinates of target agent $a$ (herder agent $i$), obey a continuity equation, that is,

$$\partial_t \rho^{T(H)}(x,t) = \nabla \cdot j^{T(H)}(x,t) + D\nabla^2 \rho^{T(H)}(x,t) \quad (11)$$

where $\nabla = \partial/\partial x$ and the currents $j^{T(H)}(x,t)$ take into account the various force terms in Eqs. (7), (8).

The (piecewise linear) pair forces comprising the (SR or LR) repulsion in Eqs. (7) and (8) can be handled straightforwardly[12] (see Supplementary Information Section 2.A for details). Here we sketch the main idea. Each of the pair terms is of the form $F_i = \sum_{j=1}^{M} F_{ij}$, where $M$ is the number of interacting neighbors. Under a mean field assumption, $F_i$ gives rise to an average force that, when neglecting nontrivial spatio-temporal correlations, has the form $\langle F(x,t) \rangle = \int F(x,y)\rho(y,t) \, dy$ where the integral extends over the interaction zone around $x$. For both the SR and LR forces, $F(x,y)$ represents a translationally invariant kernel, i.e., $F(x,y) = F(x-y)$. The resulting contribution to the current then has the form $-\langle F \rangle(x,t)\rho(x,t)$. To compute the integrals in $\langle F \rangle$, we perform a gradient expansion of $\rho(y,t)$ around $x$, i.e., $\rho(y) = \rho(x) + \nabla\rho(x)(y-x) + \mathcal{O}(\nabla\rho^2)$. Zeroth-order terms vanish due to translational invariance of the kernel. Keeping only linear terms in the gradients, we obtain the following currents stemming from repulsive forces

$$j_{\text{rep}}^T(x,t) = \rho^T(x,t)(\alpha^{\text{SR}}\nabla\rho^{\text{tot}}(x,t) + \alpha^{\text{LR}}\nabla\rho^H(x,t))$$
$$j_{\text{rep}}^H(x,t) = \rho^H(x,t)\alpha^{\text{SR}}\nabla\rho^{\text{tot}}(x,t) \quad (12)$$

where $\rho^{\text{tot}}(x,t) = \rho^T(x,t) + \rho^H(x,t)$ is the combined density field, and $\alpha^{\text{SR}} = k^{\text{SR}}\sigma^3/3$ and $\alpha^{\text{LR}} = k^T\lambda^3/3$ are positive constants related to the respective interaction radii and coupling constants in Eqs. (7), (8).

With careful consideration, the third term on the right-hand side of Eq. (8), which describes herders' decision-making abilities, can also be handled using a mean-field-like approach and a density gradient expansion. This term depends on two key components defined in previous equations: the feedback control input $\mathbf{u}_i$ [Eq. (2)] and the position of the selected target $\mathbf{T}_i^*$ [Eq. (1)]. Unlike typical classical systems where forces have a pairwise character, the resulting force here represents a three-body coupling between the herder's position, the selected target's position, and the origin. This leads to a mean-field force $\langle F(x,t) \rangle = \int F(x,y)\rho(y,t) \, dy$ with a kernel that lacks translational invariance and depends nonlinearly on both coordinates $x$ and $y$. To

proceed, we (i) linearize the (exponential) weight function appearing in the numerator of Eq. (2) (valid for $\gamma \lesssim 1/\xi$), and (ii) neglect the denominator. The kernel can then be expressed as

$$F(x,y) = -k^H(1 + \gamma(|y| - |x|))(x - (y + \text{sign}(y)\delta)) \quad (13)$$

This simplified form where the selection rule has been linearized in $x$ and $y$, still lacks translational invariance. A notable consequence is that even the zeroth-order term in the gradient expansion of $\rho^T(y)$ around $\rho^T(x)$ contributes to the average force. Combining this with the first-order (linear) gradient term, we obtain the following expression for finite values of the decision-making parameters $\gamma$ and $\delta$:

$$j_{\text{dm}}^H(x,t) = -g_1(x;\gamma,\delta)\rho^H(x,t)\rho^T(x,t) - g_2(x;\gamma,\delta)\rho^H(x,t)\nabla\rho^T(x,t). \quad (14)$$

Here, $g_{1/2}(x;\gamma,\delta)$ are space-dependent functions whose explicit expression is given by

$$g_1(\gamma,\delta,x) = \text{sign}(x)k^H \begin{cases} 2\delta\xi + \frac{2}{3}\gamma\xi^3, & \xi \leq |x| \leq L/2 - \xi \\ \left[\delta(1 - 2\gamma|x|)(|x| - \xi) + \delta(|x| + \xi) + \gamma x(\xi^2 - x^2) + \frac{2}{3}\gamma|x|^3\right], & |x| < \xi \\ \left[\delta(1 - 2\gamma(L/2 - |x|))((L/2 - |x|) - \xi) + \delta((L/2 - |x|) + \xi) \right. \\ \left. + \gamma(L/2 - |x|)(\xi^2 - (L/2 - |x|)^2) + \frac{2}{3}\gamma(L/2 - |x|)^3\right], & |x| > L/2 - \xi \end{cases} \quad (15)$$

$$g_2(\gamma,\delta,x) = k^H \begin{cases} \frac{2}{3}(\delta\gamma + 1)\xi^3, & \xi \leq |x| \leq L/2 - \xi \\ \left[\delta(1 - \gamma|x|)(\xi^2 - x^2) + \frac{2\xi^3}{3}(\delta\gamma + 1) - \frac{2}{3}(\gamma|x|)(\xi^3 - |x|^3) + (\frac{\gamma}{2})(\xi^4 - x^4)\right], & |x| < \xi \\ \left[\delta(1 - \gamma(L/2 - |x|))(\xi^2 - (L/2 - |x|)^2) + \frac{2\xi^3}{3}(\delta\gamma + 1) \right. \\ \left. - \frac{2}{3}(\gamma(L/2 - |x|))(\xi^3 - (L/2 - |x|)^3) + (\frac{\gamma}{2})(\xi^4 - (L/2 - |x|)^4)\right], & |x| > L/2 - \xi \end{cases} \quad (16)$$

Note that the functions $\upsilon_1$ and $\upsilon_2$ in (4) are scaled versions of $g_1$ and $g_2$; furthermore, to obtain $\upsilon_2$, we have to subtract from $g_2$ the (constant) contribution arising from the SR repulsion between herders and targets $\alpha^{\text{SR}}$ (see below). For further details on the derivation of $j_{\text{dm}}^H(x,t)$ see Supplementary Information Section II.A. Figure 2(a) illustrates the behaviour of $\upsilon_1$ and $\upsilon_2$. The consistently positive values of $\upsilon_2$ across all positions demonstrate the attractive force that targets exert on the herders. In contrast to $\upsilon_2$, function $\upsilon_1$ changes its sign at $x = 0$, with positive (negative) values for positive (negative) values of $x$. The resulting current acting on the herders thus depends on where the targets are relative to the goal region, consistent with our decision-making strategy. We stress that such a term is absent in more conventional nonreciprocal field theories of mixtures, e.g., the Cahn-Hillard model[12].

In the special case $\gamma = \delta = 0$ (no target selection and no trajectory planning) these functions reduce to the constants $g_1 = 0$ and $g_2 = g_2^0$ with $g_2^0 = 2k^H\xi^3/3 > 0$, yielding the current

$$j_{\text{dm}}^H(x,t)|_{\gamma = \delta = 0} = -g_2^0\rho^H(x,t)\nabla\rho^T(x,t) \quad (17)$$

Note that, even in this uncontrolled case, where herders are unable to make decisions, the resulting field equations (11) for the two density field are nonreciprocal due to the different signs and values of the cross coupling terms, as are the corresponding Langevin equations (7), (8) in this limit.

Finally, inserting the currents (12) and (14) into the continuity equations (11) we obtain the PDEs

$$\partial_t \rho^T = \nabla \cdot \left[\rho^T(\alpha^{\text{SR}}\nabla\rho^{\text{tot}} + \alpha^{\text{LR}}\nabla\rho^H)\right] + D\nabla^2\rho^T \quad (18)$$

and

$$\partial_t \rho^H = \nabla \cdot \left[ \rho^H \alpha^{SR} \nabla \rho^{tot} - g_1(x; \gamma, \delta) \rho^T \rho^H - g_2(x; \gamma, \delta) \rho^H \nabla \rho^T \right] + D \nabla^2 \rho^H \quad (19)$$

where we have omitted the arguments of the density fields. We non-dimensionalize these equations by choosing the Brownian time, $\tau \equiv \sigma^2/D$, as characteristic time scale and the particle diameter, $\sigma$, as characteristic length scale. Densities are scaled with the overall density $\rho_0 = (N_T + N_H)/L$. We further introduce dimensionless and renormalized diffusion coefficients $D^A(\rho^A) = D \frac{\tau}{\sigma^2} + \alpha^{SR} \frac{\tau \rho_0}{\sigma^2} \rho^A$ with A = {T, H}, which describe both, the effect of the noise and the intra-species short-range (SR) repulsion. With $\tilde{k}^T \equiv (\alpha^{LR} + \alpha^{SR}) \frac{\tau \rho_0}{\sigma^2}$, $v_1 \equiv g_1 \frac{\tau \rho_0}{\sigma}$, and $v_2 = (g_2 - \alpha^{SR}) \frac{\tau \rho_0}{\sigma^2}$ (and $v_2^0 = (g_2^0 - \alpha^{SR}) \frac{\tau \rho_0}{\sigma^2}$), we then arrive at the field equations given in (3), (4) and (5).

## Numerical calculations in the continuum
All simulations were carried out in MATLAB. The partial differential equations are integrated using a pseudospectral code combined with an operator splitting technique[84], which allows to accurately treat the linear part of the spatial operator (the linear diffusion). The non-linear parts of the spatial operator (non reciprocal interactions and reciprocal SR repulsion) are treated as source terms: at every time-step, they are evaluated in real space using the values from the previous step. Then they are transformed into the Fourier space, where they are considered as source terms and integrated using a fourth-order Runge-Kutta time integration scheme.

In the actual calculations, the periodically repeated, one-dimensional segment $[-L/2, L/2] \subset \mathbb{R}$ with $L = 80\sigma$ is divided into $N = 201$ equispaced grid points. The time step is chosen as $dt = 1 \cdot 10^{-4}\tau$. The simulations are run for a total duration of $t = 150\tau$, which is sufficient to reach a steady state. For Fig. 5(h), $dt = 1 \cdot 10^{-3}\tau$, $t = 750\tau$. The initial conditions are assumed to be slightly perturbed disordered states with $\rho_0^T = \rho_0^H = \rho_0/2$, where $\rho_0 = 0.5/\sigma$.

In our continuum model, some parameters can be directly adopted from the considered agent-based simulation parameters. Exceptions are, first, the sensing radius $\xi$ which we set to $\xi = 2.5\sigma$ to mitigate the effects of neglecting the denominator in Eq. (1). Second, the diffusion constant $D$ is set to $D = 5$ to increase numerical stability (notice that this changes the time scale $\tau$). Third, the amplitude of the short-range repulsion is set to $k^{SR} = 15/\tau$ so that $v_2^0 > 0$. To reproduce Fig. 2(b, d) we set $\gamma = 2.5/\sigma$, $\delta = \lambda/2$. To reproduce Fig. 5(i–l), we set $v_2(x) = \tilde{v}_2 = const$, while $v_1(x)$ is described by a square wave with different periods and amplitude $\bar{v}_1$ (details are given in the Supplementary Information Section III.A). Specifically, we set $\bar{v}_1 = k^H (2\delta\xi + (2/3)\gamma(\xi^3))(\sigma/D)\rho_0$ and $\bar{v}_2 = [k^H (2/3)(\delta\gamma + 1)(\xi^3) - \alpha^{SR}] (\rho_0/D)$, which would be the (constant) absolute values of $v_1(x)$ and $v_2(x)$ in the range $\xi \leq |x| \leq L/2 - \xi$; within these expressions, we set $\gamma = 2.5/\sigma$, $\delta = \lambda/2$. Only to reproduce Fig. 5(h) we use $k^T = 4/\tau$, while for remaining part of the manuscript, $k^T = k^H = 0.6/\tau$.

The order parameters are calculated in analogy to the corresponding agent-based calculations, as discussed previously in this Section. For the quantity $\chi(R^*)$ in Fig. 4(c), we use the definition $\chi(R^*) = \frac{1}{\mathcal{C}} \int_{-R}^{R} \rho^T(x) \, dx$, with $\mathcal{C}$ being a proper normalization factor, and $R^*$ being the value of the fitting parameter $R$ obtained from the steady state target density profile for the maximum value of $\gamma$ and $\delta$.

## Data availability
The authors declare that the data supporting the findings of this study are available within the paper and its Supplementary Information files or from the corresponding authors on request.

## Code availability
The code used for agent-based simulations and continuum field theory calculations is available at bit.ly/43BdAQi or from the corresponding authors on request.

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

## Acknowledgements

M.d.B. and A.L. wish to acknowledge support from the MUR (Italian Ministry of University and Research) Research project "Machine-learning based control of complex multi-agent systems for search and rescue operations in natural disasters" (MENTOR)—[PRIN 2022—CUP: E53D23001160006—SETTORE ERC PE7] and funding from the Scuola Superiore Meridionale in Naples, Italy which supported AL visit to TUB.

## Author contributions

S.K., M.d.B and A.L. jointly conceived and designed the research; A.L. wrote and checked the numerical code, carried out the derivations and data analysis with inputs from S.K. and M.d.B. All authors contributed to the analysis and interpretation of the results and were involved with the writing of the manuscript.

## Funding

## Competing interests

The authors declare no competing interests.
