## [Transparent Peer Review file · Nature Communications]

Nonreciprocal field theory for decision-making in multi-agent control systems

Corresponding Author: Professor Sabine Klapp

Version 0:

Reviewer comments:

Reviewer #2

(Remarks to the Author)

This paper develops a nonreciprocal field theory of herding control, extending an agent-based framework with herder and target agents to a continuum of herders and targets. Using this new field theory, the authors illustrate how a variety of collective behaviors can be recovered by variation of just two parameters and two space-dependent functions. The parameters γ and δ tune the decision-making of agents, specifically the spatial choice of target agent to influence and distance of approach of selected target. Then the functions v_1 and v_2 tune the speed distribution and the attraction/repulsion spatial distribution of targets. The authors illustrate that solutions like homogeneous distributions, confinement, expulsion, and static and travelling spatial patterns can all be recovered in the continuum limit, and are in agreement with the emergent collective behaviors observed in the agent-based formulation of the herding control problem. The framework introduced by this work is both descriptive providing a promising new tool for understanding emergent properties of collective systems across biology, social science; and prescriptive, providing a tunable tool for the design of collective behaviors in engineered groups like robotic teams. The model derivation, analysis, and simulations presented in this work are thorough and convincing. The paper is very well-written and easy to follow for an interdisciplinary audience. Overall I believe this work is a significant interdisciplinary contribution to the study of collective behavior, synthesizing agent-based control perspective with continuum perspective more commonly derived in the context of statistical physics approaches to complex systems. The results give new and broadly relevant insights into how decision-making shapes collective behavior.

I have a few minor suggestions and clarifying questions for the authors to improve the presentation of their manuscript:

1. On p.3 when authors first mention that they will be using a continuous approximation of the herding selection mechanism, it would be helpful if they were more specific (i.e. mentioned that the number of herders and the number of targets would be taken to the continuum limit) - it is not clear what is being taken to the continuum limit at this stage of the manuscript. It would also be helpful if the authors commented how the presented results are then distinct from reference 37, cited here to motivate the continuum limit.
2. In all of the figures, it consistently threw me off that the labels (a) (b) (c) etc. are displayed on the top right instead of the top left.
3. Figure 2 (e)-(g) is slightly difficult to interpret; e.g. is x increasing right to left? These plots are meant to illustrate effects of v_1 and v_2 functions, it would be useful to incorporate the mention of v_1 and v_2 directly into the image somehow to make that connection.
4. When inverse width quantity σ/Δ is introduced on p.10, it would be helpful to remind the reader that σ is the repulsion distance (Δ was introduced much more closely in the text, and the reader may forget the meaning of σ by that point)
5. The statement on p.2 "Instead, we model the targets as non-interacting Brownian particles that only experience physical volume exclusion between themselves and are repelled by herders within a distance λ " is not fully consistent with the statement on p.4 "our model includes...a reciprocal soft repulsion term between all agents within a distance σ representing the agents' diameter"; the authors should rephrase or clarify

Reviewer #3

(Remarks to the Author)

In this paper the authors introduce a microscopic Langevin equations model for herding and derive the hydrodynamic

field equations for the density field of the targets and herding agents. The motivation is clear and interesting; providing a general framework for distributed control at the large scale is both ambitious and original. The paper is well written and well illustrated and I believe accessible to a rather large audience.

Altogether, I thus recommend the paper for publication, provided the authors consider the following comments.

There are many sentences that aim at convincing the reader that the approach is very general and should generalize to a broad class of distributed learning. This is however not so clear, there are important hypothesis made, both in the microscopic model and in the derivation, which are likely to reduce this generality. One example amongst other: the form taken by the continuum equations obtained here is strongly inherited from the broken space invariance symmetry as properly stressed by the authors. For sure this does not cover all distributed control problems.

In fig 3, it is shown that for $\gamma = 0$ and $\delta = 0$, the targets and herds are mixed and the density are homogeneous, when averaged over many realizations. Conversely when $\gamma > 0$ and $\delta > 0$, herding takes place very clearly. More surprising is the fact that some level of herding is obtained when either $\gamma = 0$ or $\delta = 0$. In the previous sections, γ and δ are presented as the two necessary ingredients for distributed control to take place: a good choice of target and a good action on the target. This is a bit contradictory and the authors should explain this better.

Lines 190 to 193 : the authors write « successfully reproduces the essential features of agent-based density distributions. » and « revealing a striking similarity with their agent-based counterparts » The authors should not oversell their results. Facing fig 4, one is instead a bit disappointed by the output of the continuum theory. In short it predicts the disordered state for $\gamma = 0$ and $\delta = 0$ and the absence of homogeneous steady state for nonzero γ and δ . However it fails at identifying the regions of high shepherding. Even worst, for γ of order one, the dependence of the shepherding intensity on δ predicted by the hydrodynamics equation goes the wrong way. These limitations are not problem per se, but they should invite the authors of being more cautious about their general statements.

In the (high quality) supplementary material, the authors consider the case when there is no decision-making and with no surprise reach coupled diffusion equations akin to the non-reciprocal Cahn-Hilliard model. The linear analysis however reveals the absence of travelling wave, in contrast with what has been observed in non reciprocal field theories of phase separating system. Could the author explain why and what is the ingredient missing here for such generic travelling patterns to exist?

Minor :

Line 59 : I would rephrase the sentence : « non interacting particles that only experience physical volume exclusion » sounds a bit contradictory

Eq (2) : why not removing the minus sign in front and inverting the two terms in the bracket?

Fig 2 : Please invert the x axis and make it run between 0 and 40, instead of -40 and 0. This will be more consistent with Fig 3.

Fig 2 : please label the time axis on fig 2(b)

Eq (8) the equal sign is missing

Eq (9) d_{ij} should read d_{ai}

Line 347 : if $x > 1$ should read if $x > 0$

Eq (12) and (16) : the long range term should have a gradient of ρ^H instead of ρ^T

These are important mistakes : a careful rereading of the equations should be done before resubmission.

Version 1:

Reviewer comments:

Reviewer #2

(Remarks to the Author)

My previous review of this manuscript was positive, with minor constructive suggestions for the authors. I am satisfied with the authors' revisions in response to my previous comments. Overall I believe this is a well-written and accessible manuscript that makes a significant, unique contribution at the intersection of multi-agent control and non-reciprocal field theory. I particularly appreciate the authors' care in expanding their discussion of the derivation of the continuum equations, as well as their thorough revisions of the text and the figures for clarity and consistency of presentation in this revision of the manuscript.

Reviewer #3

(Remarks to the Author)

The authors have carefully addressed all my concerns.

I therefore recommend publication of the paper in its present form.

Response to Reviewers

A. Lama, M. di Bernardo, S. Klapp

May 31, 2025

RESPONSE TO REVIEWER #2

We thank the reviewer for their positive assessment of our work and for recognizing its interdisciplinary contributions to the study of collective behavior of decision-making systems.

1. *On p.3 when authors first mention that they will be using a continuous approximation of the shepherd-ing selection mechanism, it would be helpful if they were more specific (i.e. mentioned that the number of herders and the number of targets would be taken to the continuum limit) - it is not clear what is being taken to the continuum limit at this stage of the manuscript. It would also be helpful if the authors commented how the presented results are then distinct from reference 37, cited here to motivate the continuum limit.*

We thank the reviewer for this valuable observation. Referring to the expression in Eq. (1) as a "continuous approximation" of the selection rule can lead to confusion: rather than a continuous approximation, what we present in Eq. (1) is an approximation which allows us to perform the calculations when deriving the explicit continuum equations. The approximation is however unrelated to the continuum limit.

To clarify, we now drop the "continuous" when referring to the approximation introduced in Eq. (1), and we have revised the text on p.3 as follows:

"To facilitate the later derivation of a continuum description that explicitly incorporates decision-making, we employ an approximation of this selection mechanism that was first proposed in [1]. Specifically, we express the position of the selected target \mathbf{T}_i^* as a weighted average of the position of the targets inside the sensing region of herder i . The approximation reads

$$\mathbf{T}_i^* = \frac{\sum_{a \in N_{i,\xi}} e^{\gamma(|\mathbf{T}_a| - |\mathbf{H}_i|)} \mathbf{T}_a}{\sum_{a \in N_{i,\xi}} e^{\gamma(|\mathbf{T}_a| - |\mathbf{H}_i|)}}, \quad (1)$$

where $N_{i,\xi}$ represents the set of targets within the sensing region of the i -th herder, \mathbf{T}_a and \mathbf{H}_i are the two-dimensional (2D) Cartesian coordinates of targets and herders respectively, and $\gamma \geq 0$ is a parameter that controls the selection specificity."

Additionally, we have clarified how our work extends beyond reference 37 by adding: "We wish to highlight that the selection rule approximation given by (1) for continuum descriptions was first introduced in [1] but its application was limited to stationary target distributions, with no explicit form derived for the resulting continuum dynamics. Our work significantly extends the results presented in [1] by developing a complete and explicit theoretical framework for dynamic multi-agent systems with decision-making capabilities. "

2. *In all of the figures, it consistently threw me off that the labels (a) (b) (c) etc. are displayed on the top right instead of the top left.*

We appreciate this observation and have repositioned all figure labels to the top left corner for consistency with standard convention. This change has been implemented across all figures in the manuscript.

3. *Figure 2 (e)-(g) is slightly difficult to interpret; e.g. is x increasing right to left? These plots are meant to illustrate effects of v_1 and v_2 functions, it would be useful to incorporate the mention of v_1 and v_2 directly into the image somehow to make that connection.*

We thank the reviewer for this constructive feedback. We have comprehensively redesigned Figure 2(e)-(g) to improve clarity. The x -axis now consistently increases from left to right, and we have incorporated direct labels for the v_1 and v_2 functions within each panel. Specifically, we have added clear annotations that indicate " v_1 effect" and " v_2 effect" in the relevant plots. This redesign makes the relationship between the functions and their effects much more explicit and intuitive.

4. When inverse width quantity σ/Δ is introduced on p.10, it would be helpful to remind the reader that σ is the repulsion distance (Δ was introduced much more closely in the text, and the reader may forget the meaning of σ by that point)

We have addressed this suggestion by adding a reminder about the meaning of σ when introducing the inverse width quantity σ/Δ on p.10. The revised text now reads:

“Results for both the levels of description for χ and the inverse width (“sharpness”) σ/Δ (where σ is the repulsion distance representing the agents’ physical size) as functions of the control parameters γ and δ are shown in...”

5. The statement on p.2 “Instead, we model the targets as non-interacting Brownian particles that only experience physical volume exclusion between themselves and are repelled by herders within a distance λ ” is not fully consistent with the statement on p.4 “our model includes...a reciprocal soft repulsion term between all agents within a distance σ representing the agents’ diameter”; the authors should rephrase or clarify

We thank the reviewer for identifying this inconsistency. We have revised the statement on p.2 to ensure consistency with our model description on p.4. The corrected text now reads:

“We model the targets as Brownian particles that experience a physical (and, thus, reciprocal) volume exclusion between themselves (and with the herders) through soft repulsion within a distance σ , and are repelled by herders within a distance λ .”

This revision accurately reflects the interactions included in our mathematical model.

RESPONSE TO REVIEWER #3

We sincerely thank the reviewer for their positive assessment of our work’s clarity, originality, and accessibility. We appreciate the recommendation for publication and have carefully addressed all the comments provided.

1. There are many sentences that aim at convincing the reader that the approach is very general and should generalize to a broad class of distributed learning. This is however not so clear, there are important hypothesis made, both in the microscopic model and in the derivation, which are likely to reduce this generality. One example amongst other: the form taken by the continuum equations obtained here is strongly inherited from the broken space invariance symmetry as properly stressed by the authors. For sure this does not cover all distributed control problems.

We appreciate this important observation and have revised our claims about generality throughout the manuscript to make them clearer as suggested by the reviewer. Regarding the assumptions performed in the microscopic model, we now more explicitly acknowledge the specific hypotheses and symmetry-breaking assumptions that constrain our framework. In the introduction, we have revised the text as follows:

“Our approach does not only successfully describes the shepherding dynamics, including the transition from homogeneous to configurations where targets are confined, but also provides a general framework for incorporating decision-making in the presence of a pre-defined control goal into continuum theories of collective behavior, allowing us to reproduce a variety of collective states. In our system, the pre-defined goal corresponds to a spatially fixed region at the center of the system. This results in an “external field” that breaks translational invariance: an aspect that, as shown in the paper, is of key importance for the structure of the resulting field equations. The presence of a predefined goal region is indeed a common feature in a variety of distributed control problems [2], from bacterial systems to robotic swarm coordination [3], crowd management and autonomous transportation systems. As such, our framework opens new perspectives for analyzing and designing distributed control strategies across diverse fields of applications.”

In the discussion section (line 295), we have clarified: “agents’ decision-making capabilities inherently generate both nonreciprocity and, in presence of a pre-defined goal, entirely new forms of coupling in their continuum description...”

And at line 298: “The coupling we derive, which originates from feedback control at the microscopic level (2) and involves a goal that, in essence, breaks translational invariance, manifests as shepherding...”

In the discussion section (line 331), we have further elaborated: “The generality of our framework, is necessarily bounded by the specific hypotheses made in our microscopic model and derivation. Clearly, there are a number of problems in which decision-making occurs without broken translational invariance, such as in consensus finding [4], pattern formation in swarms [5], and various socio- and economic problems that have been very recently analyzed through the lens of a field theory [6, 7]. Still, our framework, which incorporates decision-making on the continuum level through an appropriately defined feedback current, may have relevance also for these more general decision-making problems.”

Regarding the quantitative limitations of the continuum description, we have added explicit discussion in Section II.C about how these limitations arise from the assumptions in our derivation, particularly the linearization of the selection rule (see our response to Point 3 below for details).

2. In fig 3, it is shown that for gamma = 0 and delta = 0, the targets and herds are mixed and the density are homogeneous, when averaged over many realizations. Conversely when $\gamma > 0$ and $\delta > 0$, herding takes place very clearly. More surprising is the fact that some level of herding is obtained when either $\gamma = 0$ or $\delta = 0$. In the previous sections, γ and δ are presented as the two necessary ingredients for distributed control to take place: a good choice of target and a good action on the target. This is a bit contradictory and the authors should explain this better.

This is an excellent point that deserves a more comprehensive explanation. We have expanded our discussion of the partial herding effects observed when only one decision-making parameter is active, and we also now explicitly stress that the best shepherding performance is extremely relevant for real scenarios such as swarm robotics application.

The revised text discussing Figure 3:

“The situation changes when one or both of the control parameters γ , δ are nonzero: the system develops inhomogeneities, as illustrated by the density profiles in Figs. 3(a), (b), (d). This behavior reflects a *radial symmetry-breaking* (relative to the case $\gamma = \delta = 0$) (see Supplementary Information Section II.B), where the target density $\hat{\rho}^T(r)$ reaches its maximum at the center and continuously decreases to zero radially outward. The most pronounced effect, which is key to obtain in real life scenarios such as in swarm robotics applications, occurs when γ is large compared to the sensing radius and δ is non-zero, enabling herders to effectively guide their targets toward the desired direction. We then observe a concentrated disk of targets around the center, bounded by a sharp agglomeration of herders [cf. Fig. 3(b)].

Interestingly, even when only one control parameter is nonzero, similar but less pronounced inhomogeneities emerge [Figs. 3(a, d)]. When $\gamma = 0$ but $\delta > 0$, herders are attracted to the local center of mass of targets (rather than to a specific target), but approach them with an optimized distance strategy; this leads the herders to position themselves at the back of the barycenter of the observed targets, eventually creating an accumulation of herders around the targets. Conversely, when $\delta = 0$ but $\gamma > 0$, herders approach targets from random directions but intelligently select optimal targets; this leads the herders to move towards the observed target with the largest distance from the origin, eventually generating a spatial inhomogeneity.”

3. *Lines 190 to 193 : the authors write "successfully reproduces the essential features of agent-based density distributions." and "revealing a striking similarity with their agent-based counterparts" The authors should not oversell their results. Facing fig 4, one is instead a bit disappointed by the output of the continuum theory. In short it predicts the disordered state for $\gamma = 0$ and $\delta = 0$ and the absence of homogeneous steady state for nonzero γ and δ . However it fails at identifying the regions of high shepherding. Even worst, for γ of order one, the dependence of the shepherding intensity on δ predicted by the hydrodynamics equation goes the wrong way. These limitations are not problem per se, but they should invite the authors of being more cautious about their general statements.*

We thank the reviewer for this candid assessment, which has led us to adopt a more measured tone regarding the continuum theory’s predictive capabilities. We have revised the text on lines 190-193 (starting at current line 206 in the revised manuscript) to read:

“The continuum theory captures the transition from homogeneous to spatially structured states when decision-making is present. Two exemplary long-time sets of density profiles (from the numerical solution of Eq. (3) and (4) are shown in Figs. 2(c-d), revealing the emergence of containment as we observed in their agent-based counterparts, despite differences in geometry (1D versus 2D) and specific values of γ and δ . Agent-based simulations in Supplementary Information Section I.D, where we consider a rectangular rather than circular goal region, further substantiate our claim that 1D field theories can capture the emergence of containment of the 2D agent-based dynamics. However, it is important to note that there are significant quantitative differences (for details, see Sec. 2.C).”

In section 2.C we now more clearly acknowledge the quantitative limitations of the continuum framework, and we have added “Hence, it is important to note that while our field theory predicts the absence of homogeneous steady states for nonzero γ and or δ , and thereby the key qualitative feature of the shepherding dynamics, it does not fully capture all aspects of the spatial distributions observed in agent-based simulations. These discrepancies highlight limitations in our current framework that could be addressed in future refinements, possibly by incorporating higher-order terms in the expansion of the selection rule or modifying the mean-field assumptions.”

4. *In the (high quality) supplementary material, the authors consider the case when there is no decision-making and with no surprise reach coupled diffusion equations akin to the non-reciprocal Cahn-Hilliard model. The linear analysis however reveals the absence of travelling wave, in contrast with what has been observed in non reciprocal field theories of phase separating system. Could the author explain why and what is the ingredient missing here for such generic travelling patterns to exist?*

We appreciate this insightful question about the absence of traveling waves in our model. We have added the following explanation in the supplementary material (at the end of Section II.B):

“A particularly prominent example featuring traveling patterns is the non-reciprocal Cahn-Hilliard (NRCH) model [8–10], and indeed, our present model (S21)-(S22) without decision making has important similarities with the NRCH. As in the NRCH, the present model is mass-conserving and the

cross-couplings ($\tilde{k}^T, -v_2^0$) have different signs, yielding antagonistic (i.e., anti-reciprocal) couplings between different species.

However, there is a crucial difference with our model. In the NRCH, which is a generalized model of phase separation in binary mixtures, at least one of the intraspecies couplings is negative, reflecting attractive forces within agents of the same type. This leads to phase separation and the formation of a stable interface between the two phases (realized by a finite surface tension term in the NRCH) already for reciprocal cross-couplings. The underlying phase separation in the NRCH is a crucial prerequisite for the appearance of traveling patterns at high nonreciprocity [10].

In contrast, in our model (S21)-(S22), the prefactors related to intraspecies couplings are both positive due to the underlying repulsive (excluded-volume) interactions. This precludes the occurrence of traveling patterns (see Fig. 9 in [10]). We expect, however, that such patterns could occur when we add, e.g., adhesion between the targets.”

Minor Corrections

1. Line 59 : *I would rephrase the sentence : ” non interacting particles that only experience physical volume exclusion ” sounds a bit contradictory*

We have rephrased this sentence to eliminate the contradiction. The revised text now reads: “We model the targets as Brownian particles that experience a physical (and, thus, reciprocal) volume exclusion between themselves (and with the herders) through soft repulsion within a distance σ , and are repelled by herders within a distance λ .”

2. Eq (2) : *why not removing the minus sign in front and inverting the two terms in the bracket? We have implemented this suggestion to improve clarity. Equation (2) has been reformulated by removing the minus sign and inverting the terms in the bracket.*

3. Fig 2 : *Please invert the x axis and make it run between 0 and 40, instead of -40 and 0. This will be more consistent with Fig 3.*

We have adjusted Figure 2 as suggested, with the x-axis now running from 0 to 40, which provides better consistency with Figure 3.

4. Fig 2 : *please label the time axis on fig 2(b)*

We have added the missing time axis label to Figure 2(b), now reading t/τ ”.

5. Eq (8) *the equal sign is missing*

We have added the missing equal sign to Equation (8).

6. Eq (9) *d_{ij} should read d_{ai}*

We have corrected this error, changing d_{ij} to d_{ai} in Equation (9).

7. Line 347 : *if $x > 1$ should read if $x > 0$*

We have corrected “if $x > 1$ ” to “if $x > 0$ ” on Line 347 (now line 373).

8. Eq (12) and (16) : *the long range term should have a gradient of ρ^H instead of ρ^T*

We have fixed this important error in Equations (12) and (18), replacing $\nabla\rho^T$ with $\nabla\rho^H$ in the long-range term.

9. These are important mistakes : *a careful rereading of the equations should be done before resubmission.*

We thank the reviewer for identifying these errors. We have conducted a thorough review of all equations in the manuscript and supplementary materials to ensure no further errors remain. This comprehensive check included verifying consistency in notation, correctness of all mathematical expressions, and proper labeling throughout the document.

-
- [1] A. Lama and M. di Bernardo, Shepherding and herdability in complex multiagent systems, *Physical Review Research* **6**, L032012 (2024).
 - [2] G. F. Franklin, J. D. Powell, A. Emami-Naeini, and J. D. Powell, *Feedback control of dynamic systems*, Vol. 4 (Prentice hall Upper Saddle River, 2002).
 - [3] R. Vaughan, N. Sumpter, J. Henderson, A. Frost, and S. Cameron, Experiments in automatic flock control, *Robotics and Autonomous Systems* **31**, 109 (2000).
 - [4] T. Li, M. Fu, L. Xie, and J.-F. Zhang, Distributed consensus with limited communication data rate, *IEEE Transactions on Automatic Control* **56**, 279 (2011).
 - [5] A. Giusti, G. C. Maffettone, D. Fiore, M. Coraggio, and M. di Bernardo, Distributed control for geometric pattern formation of large-scale multirobot systems, *Frontiers in Robotics and AI* **10**, 1219931 (2023).
 - [6] D. S. Seara, J. Colen, M. Fruchart, Y. Avni, D. Martin, and V. Vitelli, *Sociohydrodynamics: data-driven modelling of social behavior* (2025), [arXiv:2312.17627 \[cond-mat.soft\]](https://arxiv.org/abs/2312.17627).
 - [7] R. Zakine, J. Garnier-Brun, A.-C. Becharat, and M. Benzaquen, Socioeconomic agents as active matter in nonequilibrium sakoda-schelling models, *Physical Review E* **109**, 044310 (2024).
 - [8] Z. You, A. Baskaran, and M. C. Marchetti, Nonreciprocity as a generic route to traveling states, *Proceedings of the National Academy of Sciences* **117**, 19767 (2020).
 - [9] S. Saha, J. Agudo-Canalejo, and R. Golestanian, Scalar active mixtures: The nonreciprocal cahn-hilliard model, *Physical Review X* **10**, 041009 (2020).
 - [10] F. Brauns and M. C. Marchetti, Nonreciprocal pattern formation of conserved fields, *Physical Review X* **14**, 021014 (2024).

Response to Reviewers

A. Lama, M. di Bernardo, S. Klapp

July 25, 2025

We thank both reviewers for their careful and positive evaluations and for the very constructive suggestions in the first round that have greatly helped to improve the manuscript. We are happy to hear that the reviewers find our revisions satisfactory and recommend publication of the manuscript in its present form. We would also like to thank the reviewers for recognizing the manuscript's interdisciplinary contributions at the intersection of multi-agent control and non-reciprocal field theory.